# Decomposed Multilateral Filtering for Accelerating Filtering with Multiple Guidance Images

**DOI:** 10.3390/s24020633

**Published:** 2024-01-19

**Authors:** Haruki Nogami, Yamato Kanetaka, Yuki Naganawa, Yoshihiro Maeda, Norishige Fukushima

**Affiliations:** 1Department of Computer Science, Faculty of Engineering, Nagoya Institute of Technology, Gokiso-cho, Showa-ku, Nagoya 466-8555, Japany.kanetaka.099@stn.nitech.ac.jp (Y.K.);; 2Department of Electrical Engineering, Faculty of Engineering, Tokyo University of Science, Tokyo 125-8585, Japan; ymaeda@rs.tus.ac.jp

**Keywords:** constant-time filtering, edge-preserving filtering, multilateral filtering

## Abstract

This paper proposes an efficient algorithm for edge-preserving filtering with multiple guidance images, so-called multilateral filtering. Multimodal signal processing for sensor fusion is increasingly important in image sensing. Edge-preserving filtering is available for various sensor fusion applications, such as estimating scene properties and refining inverse-rendered images. The main application is joint edge-preserving filtering, which can preferably reflect the edge information of a guidance image from an additional sensor. The drawback of edge-preserving filtering lies in its long computational time; thus, many acceleration methods have been proposed. However, most accelerated filtering cannot handle multiple guidance information well, although the multiple guidance information provides us with various benefits. Therefore, we extend the efficient edge-preserving filters so that they can use additional multiple guidance images. Our algorithm, named decomposes multilateral filtering (DMF), can extend the efficient filtering methods to the multilateral filtering method, which decomposes the filter into a set of constant-time filtering. Experimental results show that our algorithm performs efficiently and is sufficient for various applications.

## 1. Introduction

Multimodal signal processing for sensor fusion is increasingly important in image sensing. Sensor fusion can combine beneficial information from different sensors to generate a richer single image. Image signal fusion approaches have various applications: RGB and infrared image fusion [1,2,3], RGB and multispectral image fusion [4], intercolor RGB signal fusion [5,6], RGB and depth fusion [7,8], RGB and light fusion [9], RGB and computed edge fusion [10], different focal image fusion [11,12], CT and MRI signal fusion for medical image processing [3], Retinex-based enhancement [13], SAR and multispectral image fusion [14], and general signal fusion [15].

Filtering is a basic tool for handling such multimodal signals. Multilateral filtering, which is one type of edge-preserving filtering, successfully handles multiple signal information. Edge-preserving filtering with additional guidance information, called joint edge-preserving filtering, recently attracted attention from image processing and computational photography researchers for sensor fusion. Joint edge-preserving filtering helps transfer major characteristics from guidance images, which are not filtering images themselves. Various applications use the filters, including flash/no-flash photography [16,17], up-sampling/super resolution [18], compression noise removal [19], alpha matting [20], haze removing [21], rain removing [22], depth refinement [23,24], stereo matching [25,26], and optical flow estimation [27].

Joint/cross bilateral filtering [16,17] is a seminal work of joint edge-preserving filtering. The filter is naturally derived from bilateral filtering [28] by computing the kernel weight from a guidance image instead of an input filtering image. This formulation enables us to reflect the edge information of the guidance image (e.g., RGB, infrared, and hyperspectral images) to the filtering target image (e.g., RGB image, alpha mask, depth map, and optical flow).

We can expect a higher edge-preserving effect using multiple guidance images (e.g., a set of multi-sensor signals), and recently, we have been able to capture not only RGB images but also infrared, hyperspectral, depth, and other images by new devices (e.g., infrared/hyperspectral cameras and depth sensors). The images have different edge information and signal characteristics from RGB images. The multiple guidance information is helpful for improving signal visibility and the signal-to-noise ratio [29,30]. In other cases, we can deal with a self-generated image and inversely rendered maps as an additional guidance signal [31,32].

There are two categories for using multiple guidance images in image filtering: high-dimensional filtering [30,33,34,35,36,37] and multilateral filtering [29,31,32,38,39]. The former is additive logic, and the latter is multiplicative logic for additional kernels. The main difference is the severity of the restriction to compute the kernel weight. The restriction of additive logic is looser than that of multiplicative logic; hence, high-dimensional filtering can robustly smooth out noise or rich textures. By contrast, since the restriction of the multiplicative logic is severe, multilateral filtering produces fewer blurred regions. Each filtering method has advantages and disadvantages, but multilateral filtering is preferred when we expect a sharply edge-preserving effect.

A critical issue of edge-preserving filtering for multimodel sensing is computational time. This is because sensing is the gateway to all processing, and signal processing during sensing is expected to operate in real time. Therefore, many researchers have proposed acceleration methods for edge-preserving filtering. In particular, the acceleration for bilateral filtering has been actively discussed. The bilateral grid [40,41] is the seminal approach, and Yang et al. [42,43] extend it to bilateral filtering in constant time. Yang’s method [43] has adequate efficiency in grayscale images, and recent work further accelerates the bilateral filter [44,45,46]; however, they are inefficient in color cases. There are several proposals [34,47,48] to approximate and accelerate bilateral filtering in the case of higher-dimensional (color) images. Furthermore, hardware-friendly methods are proposed [49,50,51]. However, these approaches have limitations for kernel weight, whose kernels are defined by the Gaussian distribution. Other efficient edge-preserving filters, which do not limit the Gaussian distribution, are proposed in contrast to the bilateral filtering acceleration. Guided image filtering [20], domain transform filtering [33], and adaptive manifold filtering [30] are representative examples. These filters have assumptions different from Gaussian smoothing but have excellent edge-preserving effects and efficiency. Note that these filters can handle similar signals better than those with different modalities and characteristics.

Multiple guidance images provide richer information for various applications; however, these efficient methods cannot individually handle multiple guidance images. Therefore, we propose an efficient algorithm for accelerating multilateral filtering, which is developed for multiple-guidance image filtering. Furthermore, we extend the efficient edge-preserving filters so that they can exploit multiple guidance images.

Our algorithm is based on the fact that *n*-lateral filtering is represented by the summation of (n−1)-lateral filtering. Therefore, when multilateral filtering is expanded as an asymptotic expression, it becomes constant-time filtering since 1-lateral filtering is spatial filtering. Figure 1 denotes the overview of the proposed filter algorithm. The proposed filter—named DMF: decomposed multilateral filtering—recursively decomposes multilateral (*n*-lateral) filtering by splatting to (n−1)-lateral filtering until it is a constant-time filter. Then, the results of constant-time filtering for the decomposed components are merged into the result of multilateral filtering.

The contributions of this paper are summarized as follows:1.Introducing a constant-time algorithm for multilateral filtering (Section 5);2.Extending various filters (e.g., guided image filtering [20] and domain transform filtering [33]) to deal with multiple guidance information (Section 6.1);3.Proposing a multilateral extension to the filter that uses the filtering output as a guidance image, such as rolling guidance filters [52] (Section 6.2).

## 2. Related Work

Due to physical constraints, a single image sensor cannot simultaneously capture rich information such as resolution, wavelength range, focus, dynamic range, and scene features. Image fusion is one way to solve this problem. Research on image fusion is active, with the number of papers increasing each year, as well as many survey papers [53,54,55,56,57,58,59,60,61,62]. Image fusion involves smoothing, denoising, enhancement, sharpening, super-resolution, and blending for multiple signals to obtain the desired signal. Image fusion is mainly divided into digital photography image fusion and multimodal image fusion.

Digital photography image fusion combines images taken by the same sensor with different sensor settings and includes multi-focus image fusion, multi-exposure image fusion, multi-temporal image fusion, and multi-view image fusion. In multi-focus image fusion, an all-in-focus image is synthesized from images taken at different focus settings, and in multi-exposure image fusion, a wide dynamic range image is synthesized from images taken at different dynamic ranges. Multi-exposure image fusion also includes the use of different external flash environments. Multi-temporal image fusion synthesizes signals that vary along a time axis, while multi-view image fusion synthesizes signals from camera motion or multiple cameras capturing a scene.

Multimodal image fusion combines different characteristics of multiple sensors into one, including RGB-IR fusion, multi-hyperspectral-panchromatic image fusion, RGB-depth/LiDAR fusion, and medical image fusion (CT, PET, MRI, SPECT, X-ray), etc. In RGB-IR fusion, visible images are combined with IR images, taking advantage of the high contrast of IR and the good texture characteristics of RGB in the visible region. It also combines images using the different wavelength bands that can be captured by external flashes. In multi-hyperspectral-panchromatic image fusion, sensors that acquire images in different wavelength bands and resolutions are combined, and each sensor often has a different resolution and noise sensitivity. The objective is to improve the resolution and noise sensitivity of each sensor. RGB-depth/LiDAR fusion corrects depth sensor output from RGB images, including upsampling of depth information, missing interpolation, and contour correction noise reduction. Medical image fusion integrates the output of various medical sensors in the same dimension to assist in the diagnosis.

Among these image fusion methods, those that improve the acquisition signal are called sharpening fusion, which aims at signal denoising, sharpening, contrast improvement, and resolution improvement. In image fusion, various tools are used, such as weighted smoothing filtering, morphology filtering, principal component analysis (PCA), Laplacian pyramid, discrete cosine transformation (DCT), discrete Fourier transformation (DFT), discrete wavelet transform (DWT), etc. This paper is an extension of the weighted smoothing method. In particular, the proposed method extends existing smoothing/weighted smoothing methods to guided smoothing and has a wide range of applications.

## 3. Preliminaries

In this section, we review the previous work of constant-time bilateral filtering proposed by Yang et al. [42,43]. Bilateral filtering [28] is a representative edge-preserving smoothing filtering defined as a finite impulse response (FIR) manner. This filtering achieves edge-preserving effects by filtering in the range and spatial domains; thus, its filtering kernel weights are derived from a product of spatial and range weights based on a Gaussian distribution. Let input and output images be denoted as $I,O^{B}:S\to R$, where $S\subset Z^{D}$ is the spatial domain, $R={\left[0,n\;-\;1\right]}^{d}\subset R^{d}$ is the range domain, and *d* is the color range dimension (generally, D=2, N=256, and d=3), respectively. Bilateral filtering is formulated as follows: (1)$$O_{p}^{B}=\frac{\sum_{q\in N_{p}}f_{S}\left(p,q\right)f_{R}\left(I_{p},I_{q}\right)I_{q}}{\sum_{q\in N_{p}}f_{S}\left(p,q\right)f_{R}\left(I_{p},I_{q}\right)},$$
where p,q∈S represents a target pixel and a neighboring pixel of p, respectively. $I_{p},I_{q}\in R$ are pixel values at p,q. $N_{p}\subset S$ is a set of neighboring pixels of p. $f_{S}:S\times S\to R,f_{R}:R\times R\to R$ are weight functions based on the Gaussian distribution whose smoothing parameters are $\sigma_{S}$ and $\sigma_{R}$, respectively. Here, we can formulate joint bilateral filtering [16,17] by replacing *I* in (Equation 1) with an arbitrary additional guidance image J:S→R.

Naïve bilateral filtering is $O(r^{2})$ per pixel algorithm, where *r* is the filtering kernel radius; thus, the computational complexity increases exponentially when the kernel size is large. Several constant-time-per-pixel algorithms for bilateral filtering have been proposed to solve this problem. In particular, the algorithm proposed by Yang et al. [42,43] is the basis of the proposed method.

Yang et al. proposed a constant-time algorithm by extending the bilateral grid [40,41]. The algorithm decomposes bilateral filtering into a set of spatial filtering that can be computed in constant-time (e.g., box filter using integral image [63,64] and the recursive Gaussian filter [65,66,67,68]). The decomposition is conducted by computing principle bilateral filtered image components (PBFICs) [43] from the input or guidance image. Since arbitrary range filtering weights can generate PBFICs, the algorithm can compute the arbitrary bilateral filtering response in the range kernel.

Yang’s algorithm [43] is further extended to apply to multichannel images in [42]. The extended algorithm computes multichannel images by preparing multichannel PBFICs with combinations of pixel values in each channel. However, this extension requires uniform processing for all channels. In other words, we cannot filter for each channel differently. This indicates that the algorithm is extendable when we compute multichannel or multiple guidance images with differential characteristics in each channel.

Our algorithm is inspired by Yang’s algorithm [42,43], which represents bilateral filtering by a set of spatial filtering. In contrast, our algorithm decomposes a filter for multichannel images into arbitrary constant-time filters.

## 4. Relationship between Multilateral Filtering and Higher-Dimensional Filtering

In this section, we compare the filtering properties between multilateral filtering (MF) and high-dimensional filtering (HDF). The main difference between them is the logic to compute the filtering weight. The weight of multilateral filtering $f_{M}\in R$ is computed by the multiplicative logic from spatial weight and range weights of multiple guidance images: (2)$$f_{M}\left(p,q\right)=f_{S}\left(p,q\right)\prod_{i=1}^{m}f_{R}^{i}\left(J_{p}^{i},J_{q}^{i}\right),$$
where $f_{R}^{i}:R^{i}\times R^{i}\to R$ is a filtering weight for the *i*-th guidance image $J^{i}:S\to R^{i}$, where $R^{i}$ is the range domain of $J^{i}$. *m* is the number of guidance images. An early work on MF was proposed by Choudhury and Tumblin [32]. Each range weight $f_{R}^{i}$ for the guidance image is individually defined to represent the characteristics of the image.

HDF’s weight $f_{H}\in R$ is computed by the additive logic: (3)$$f_{H}\left(p,q\right)=\rho \left(\sum_{i=1}^{|V_{p}|}l_{\gamma }\left(V_{p}\left(i\right)-V_{p}\left(i\right)\right)\right),$$
where ρ∈R denotes an arbitrary weight function at the pixel p; $l_{\gamma }\in R$ denotes an arbitrary norm function; $V_{p}$ denotes higher-dimensional information consisting of spatial and range information, e.g., $V_{p}=\left(x_{p},y_{p},r_{p},g_{p},b_{p}\right)$ in RGB image, and $|V_{p}|\in Z$ is the size of $V_{p}$. The work of Gastal and Oliveira [30] is a successful extension for HDF with multiple guidance information. They exploited additional guidance information to increase higher-dimensional information V.

The two logics differ in terms of the severity of the restriction to compute the kernel weight; the multiplicative logic’s restriction is more severe than the additive logic. The difference affects the edge-preserving performance. Figure 2 shows examples of HDF and MF weights. HDF assigns the low weights as a whole, even if the guidance pixel is hardly relevant to the target pixel. In contrast, MF assigns the low weights with the guidance pixel having a similar target pixel value. In this way, MF has a high edge preservation effect; hence, it is preferred when it is significant.

## 5. Proposed Method: Decomposed Multilateral Filtering

The proposed filter of DMF first decomposes MF until constant-time filtering. This allows us to convert the computational complexity from $O(r^{2})$ to O(1) per pixel. This section defines MF and proves its decomposability. Algorithm 1 reviews the flow of DMF. Next, we discuss the extension of the algorithm.

### 5.1. Definition of Multilateral Filtering

MF assumes that the filtering weight is derived from the multiplicative logic discussed in Section 4. Furthermore, MF is equivalent to *n*-lateral filtering when n−1 guidance images are used for filtering. When n=1 or 2, *n*-lateral filtering means spatial filtering or bilateral filtering, respectively. Therefore, we assume n≥3 in this section and compute *n*-lateral filtering output $O^{n}:S\to R$ as follows: (4)$$\begin{array}{ccc} O_{p}^{n} & =\frac{\sum_{q\in N_{p}}f^{n}\left(p,q\right)I_{q}}{\sum_{q\in N_{p}}f^{n}\left(p,q\right)}, & \end{array}$$(5)$$\begin{array}{ccc} f^{n}\left(p,q\right) & =f_{S}\left(p,q\right)\prod_{i=1}^{n\;-\;1}f_{R}^{i}\left(J_{p}^{i},J_{q}^{i}\right) & \;(n\geq 2), \end{array}$$(6)$$\begin{array}{ccc} f^{1}\left(p,q\right) & =f_{S}\left(p,q\right) & \;(n=1), \end{array}$$
where $f_{R}^{i}$ denotes the range filtering weight for *i*-th guidance image $J^{i}$. The *n*-lateral filtering weight is $f^{n}\in R$. Equations (Equation 4) and (5) are the basic formulation of MF. Here, we basically define the first filter of $f^{1}$ as a spatial filter in (6), which is an arbitrary linear-time invariant (LTI) filter (e.g., box, circler, Gaussian, and Gabor filters) and linear-time variant (LTV) filters (e.g., spatially adaptive Gaussian filter [69]). LTI filters can be performed in O(1) by sliding DCT [68] filtering, but adaptive filtering has difficulty in converting O(1) filters.
**Algorithm 1** Decomposed Multilateral Filtering**function** *n*-lateral_filtering(n,J,I)    // $J=\{J^{1},J^{2},\ldots ,J^{n-1}\}$    **for all** values *k*$\left(0\leq k\leq T^{n\;-\;1}-1\right)$ **do**        // *Splatting as Equations (Equation 11) and (12)*        $W_{L_{k}^{n\;-\;1}}\leftarrow f_{R}^{n\;-\;1}\left(k,J^{n\;-\;1}\right)I$        $K_{L_{k}^{n\;-\;1}}\leftarrow f_{R}^{n\;-\;1}\left(k,J^{n\;-\;1}\right)$        //*(n−1)-lateral filtering*
        **if** n≥2 **then**           $W_{L_{k}^{n\;-\;1}}$←*n*-lateral_filtering($n\;-\;1,J,W_{L_{k}^{n\;-\;1}}$)           $K_{L_{k}^{n\;-\;1}}$←*n*-lateral_filtering($n\;-\;1,J,K_{L_{k}^{n\;-\;1}}$)        **else**           // *Final filtering step*           $W_{L_{k}^{n\;-\;1}}$←$f_{S}*W_{L_{k}^{n\;-\;1}}$           $K_{L_{k}^{n\;-\;1}}$←$f_{S}*K_{L_{k}^{n\;-\;1}}$        **end if**        // *Normalization as Equation (Equation 13)*        $C_{L_{k}^{n\;-\;1}}$←$K_{L_{k}^{n\;-\;1}}/W_{L_{k}^{n\;-\;1}}$    **end for**    // *Interpolation as Equation (Equation 15)*    //$L^{n\;-\;1}=\{L_{1}^{n\;-\;1},\ldots ,L_{T^{n\;-\;1}\;-\;1}^{n\;-\;1}$}    $O^{n}$← Interpolation($J^{n\;-\;1},L^{n\;-\;1},C_{L^{n\;-\;1}}$)    **return** $O^{n}$**end function**

### 5.2. Recursive Representation for Decomposed Multilateral Filtering

We introduce DMF and prove the decomposability of MF in this subsection. In (5), the *n*-lateral filtering weight $f^{n}$ can be replaced with the product of its one-dimensional lower weight $f^{n\;-\;1}$ and the range weight $f_{R}^{n\;-\;1}$: (7)$$f^{n}\left(p,q\right)=f^{n\;-\;1}\left(p,q\right)f_{R}^{n\;-\;1}\left(J_{p}^{n\;-\;1},J_{q}^{n\;-\;1}\right).$$

We can re-formulate MF from (Equation 4) using (Equation 7) as follows: (8)$$O_{p}^{n}=\frac{\sum_{q\in N_{p}}f^{n\;-\;1}\left(p,q\right)f_{R}^{n\;-\;1}\left(J_{p}^{n\;-\;1},J_{q}^{n\;-\;1}\right)I_{q}}{\sum_{q\in N_{p}}f^{n\;-\;1}\left(p,q\right)f_{R}^{n\;-\;1}\left(J_{p}^{n\;-\;1},J_{q}^{n\;-\;1}\right)}.$$
This form shows that we can express *n*-lateral filtering using (n−1)-lateral filtering weight. Furthermore, we deform (Equation 8) by the additional assumption that the pixel values of the guidance images are discrete. Let $c\in R^{n\;-\;1}=\left[0,N^{n\;-\;1}-1\right]$ be a constant value, where $N^{n\;-\;1}$ is the number of tones in the range of the (n−1)-th guidance image. When a pixel value in the *n*-th guidance image $J_{p}^{n\;-\;1}$ in (Equation 8) is replaced by a constant value *c*, it is rewritten as follows: (9)$$\begin{array}{cc} C_{c,p}^{n\;-\;1} & =\frac{\sum_{q\in N_{p}}f^{n\;-\;1}\left(p,q\right)f_{R}^{n\;-\;1}\left(c,J_{q}^{n\;-\;1}\right)I_{q}}{\sum_{q\in N_{p}}f^{n\;-\;1}\left(p,q\right)f_{R}^{n\;-\;1}\left(c,J_{q}^{n\;-\;1}\right)} \end{array}$$(10)$$\begin{array}{cc} O_{p}^{n} & =C_{v,p}^{n\;-\;1}\;s.t.\;v=\underset{x}{\arg\;\min}{∥x-J^{n\;-\;1}∥}_{1}, \end{array}$$
where ${∥\cdot ∥}_{1}$ is $l_{1}$ norm operator. We call $C_{c}^{n\;-\;1}:S\to R$ a component image of *n*-lateral filtering, and its pixel value at p is denoted by $C_{c,p}^{n\;-\;1}\in R$.

$f_{R}^{n\;-\;1}\left(c,J_{q}^{n\;-\;1}\right)$ and $f_{R}^{n\;-\;1}\left(c,J_{q}^{n\;-\;1}\right)I_{q}$ in (Equation 9) can be cached as images in constant-time; hence, we express these coefficients as the following images: (11)$$\begin{array}{cc} W_{c,q}^{n\;-\;1} & =f_{R}^{n\;-\;1}\left(c,J_{q}^{n\;-\;1}\right) \end{array}$$(12)$$\begin{array}{cc} K_{c,q}^{n\;-\;1} & =W_{c,q}^{n\;-\;1}I_{q}, \end{array}$$
where $W_{c}^{n\;-\;1}:S\to R^{n\;-\;1}$ and $K_{c}^{n\;-\;1}:S\to R$ are the elements of the denominator and numerator in (Equation 9), respectively. We call the processes for Equations (Equation 11) and (12) as splatting followed by the paper [47,48], and we call the images as coefficient image. For simplification, we rewrite (Equation 9) using these coefficient images and the convolution operator ∗: (13)$$C_{c}^{n\;-\;1}=\frac{f^{n\;-\;1}*K_{c}^{n\;-\;1}}{f^{n\;-\;1}*W_{c}^{n\;-\;1}},$$
where the pixel operator p can be dropped. Figure 3 shows the splatting procedure in DMF.

The denominator and numerator in (Equation 13) represent (n−1)-lateral filtering. This indicates that *n*-lateral filtering has been decomposed into (n−1)-lateral filtering. Therefore, MF can be decomposed recursively: (14)$$f^{n}=g^{n\;-\;1}\circ f^{n\;-\;1}=g^{n\;-\;1}\circ g^{n\;-\;2}\circ \cdots \circ g^{1}\circ f^{1},$$
where $g^{x}\circ$ denotes a decomposing operator, as described in Equations (Equation 11)–(Equation 13). Equation (Equation 14) summarizes the DMF formulation. Since $f^{1}$ (e.g., Gaussian filtering) can be computed in constant-time per pixel by recursive algorithms, and the decomposing operation is independent of kernel size, DMF can also be computed in constant-time.

### 5.3. Tonal Range Subsampling

The exact filtering result can be obtained by computing coefficient images for all values $c\in R^{n\;-\;1}$ ranged in the guidance image $J^{n\;-\;1}$. Here, we increase efficiency by quantizing the guidance tonal ranges. Let $L^{n\;-\;1}$ be a quantized set of $R^{n\;-\;1}$, and $T^{n\;-\;1}=\left|L^{n\;-\;1}\right|$ be the number of tones in a quantized tonal range of the (n−1)-guidance image, where $T^{n\;-\;1}\leq \left|R^{n\;-\;1}\right|$. Furthermore, let $L_{k}^{n\;-\;1}\in L^{n\;-\;1}$ be the *k*-th label’s value, where $k\in \{0,1,\ldots ,|T^{n\;-\;1}|\}$ that the return value is in the quantized range domain. We can obtain the final output of DMF by linear interpolation of the current and next coefficient images (i.e., $C_{L_{k}^{n\;-\;1}}^{n\;-\;1}$ and $C_{L_{k+1}^{n\;-\;1}}^{n\;-\;1}$): (15)$$O_{p}^{n}\approx \left(L_{k+1}^{n\;-\;1}-J_{p}^{n\;-\;1}\right)C_{L_{k}^{n\;-\;1},p}^{n\;-\;1}+\left(J_{p}^{n\;-\;1}-L_{k}^{n\;-\;1}\right)C_{L_{k+1}^{n\;-\;1},p}^{n\;-\;1}$$
(16)$$s.t.\;k=\underset{x\in L^{n\;-\;1}}{\arg\;\min}{∥J_{p}^{n\;-\;1}-L_{x}^{n\;-\;1}∥}_{1}.$$

### 5.4. Spatial Domain Subsampling

Considering the sparsity of the coefficient images, we can also apply subsampling in the spatial domain for further increasing efficiency, as discussed in [41,42]. In the DMF case, we can apply spatial subsampling to the coefficient images in several steps: the first and the second decomposition. If we apply spatial subsampling to DMF in the *n*-lateral filtering splatting, the process is computed as: $$\begin{array}{cc} K_{c}^{n\;-\;1↓} & =\mathrm{downsample}(K_{c}^{n\;-\;1}) \end{array}$$(17)$$\begin{array}{cc} W_{c}^{n\;-\;1↓} & =\mathrm{downsample}(W_{c}^{n\;-\;1}) \end{array}$$(18)$$\begin{array}{cc} C_{c}^{n\;-\;1↓} & =\frac{f^{n\;-\;1}*K_{c}^{n\;-\;1↓}}{f^{n\;-\;1}*W_{c}^{n\;-\;1↓}} \end{array}$$(19)$$\begin{array}{cc} C_{c}^{n\;-\;1}\approx C_{c}^{n\;-\;1↓↑} & =\mathrm{upsample}(C_{c}^{n\;-\;1↓}), \end{array}$$
where $X^{↓}$ and $X^{↓↑}$ (X={K,W,C}) are the downsampled and upsampled images, respectively. We use the average nearest-neighbor pixels and linear interpolation for subsampling and upsampling, respectively. Our method can apply different ratios of spatial subsampling to arbitrary guidance channels based on the sparsity of each channel (e.g., YUV image components of JPEG and MPEG format, RGB-D images). The flexibility is an advantage for Yang’s algorithm [42].

## 6. Extension of Decomposed Multilateral Filtering

### 6.1. Beyond Gauss Transform

DMF can deal with any multilateral filtering responses and does not limit the Gauss transform [70], which is the combination of Gaussian filtering. DMF can select arbitrary ranges and spatial filters. Furthermore, we can select the filtering responses by changing the final filtering step.

We should have DMF until the spatial filtering in (6). In contrast, DMF does not always require decomposition until it is spatial filtering. Specifically, we can apply any joint edge-preserving filters for the final filtering step while decreasing the number of decompositions. Some edge-preserving filtering can handle multichannel signals in the designed weight function. For example, high-dimensional Gaussian filtering handles multichannel signals by the Gaussian distribution with the Euclid distance; instead, domain transform filtering uses the geodesic distance. Guided image filtering handles them by the local linear model with $l_{2}$ norm between signals.

Therefore, when the final filter is performed in edge-preserving filtering, the DMF decomposition can be reduced by the number of dimensions handled by the edge-preserving filtering. Let the handling signal set be $G=\{J_{s},\ldots ,J_{1}\}$, where *s* is the number of handling channels (i.e., s=3 in the RGB image case). Using edge-preserving filtering, (Equation 13) of the final step is replaced as: (20)$$C_{c}^{s}=\frac{H_{G}*K_{c}^{s}}{H_{G}*W_{c}^{s}},$$
where $H_{G}*$ represents any joint edge-preserving filtering with the guidance signal set as G. Examples of the final step filtering are high-dimensional filtering (high-dimensional Gaussian filtering [47,48], guided image filtering [20,71], domain transform filtering [33], adaptive manifold filtering [30]), frequency transform filtering ( edge-avoiding wavelet [72,73], redundant frequency transform [74]), adaptive filtering (range parameter adaptive filtering [75]), enhancement filtering (local Laplacian filtering [76,77,78]), statical filtering (fast guided median filtering [79]), LUT-based filtering [80], optimization-based filtering (weighted least square optimization [81,82], and L0 smoothing optimization [83]).

The representation allows us to extend various edge-preserving filtering methods to handle multiple guidance images. This fact is helpful for various applications since the required filtering properties, e.g., local linearity [20] and geodesic distance [33], are different by application. Note that it has the potential to be faster because of the merged treatment of dimensions, but the filter may not work well if the characteristics of the combined set are not identical.

### 6.2. Multilateral Rolling Guidance Filtering

MF is also helpful in self-generating multiple guidance information from single guidance information, such as rolling guidance filtering [52]. Rolling guidance filtering is iteratively processed using the filtered image as the guidance image. In this regard, the filtering image is fixed as the input image and the guidance image varies. This iterative representation can be applied to multilateral filtering with some modifications. We call it multilateral rolling guidance filtering (MRGF) and show the actual processing in Figure 4. The main difference is the filter output as an additional guidance image.

MRGF is specifically practical when edge information is essential, such as image segmentation and feathering. Since we can reflect the smoothed or refined results to the filtering target image, it tends to refine the desirable features for the target image. Significantly, the first estimated maps of scene properties often contain noises and errors. MRGF has good performance in the refinement of maps. We verify the performance of MRGF in the following experimental part.

## 7. Experimental Results

We evaluated the proposed filter of DMF in terms of accuracy and efficiency. The implementations for DMF and competitive methods are written in C++, and the codes are parallelized by OpenMP and vectorized by AVX2. We used Intel Core i7 7700K (four cores, eight threads) and Visual Studio2022 compiler for experiments.

### 7.1. Accuracy Evaluation

First, we evaluated how much the DMF result corresponds to the naïve implementation of the MF result. In our experiments, we applied our algorithm to trilateral filtering [29]. Note that the Gauss transform is applied to the trilateral filtering weights for spatial and range weights, where the standard deviations are $\sigma_{s}$, $\sigma_{r1}$, and $\sigma_{r2}$. Here, $\sigma_{r1}$ and $\sigma_{r2}$ are the parameters for the tonal ranges of the guidance and filtering images, respectively. We apply recursive Gaussian filtering with sliding DCT [68,84] for spatial filtering. We evaluate the accuracy of our algorithm by flash/no-flash denoising [16,17]. The range kernels $f_{R_{1}}$ and $f_{R_{2}}$ for MF are computed from flash images and no-flash images, respectively. We use the peak signal-to-noise ratio (PSNR) [85,86] as the objective evaluation method of the approximation accuracy between naïve results and the approximation results. The evaluation formula used is as follows: (21)$$\begin{array}{c} PSNR=10\log_{10}\frac{S\cdot 255^{2}}{{∥A-G∥}_{2}^{2}}, \end{array}$$ where ${∥\cdot ∥}_{2}$ is the $l_{2}$ norm operator, **A** is approximated signals proceded by DMF, and **G** is ground truth signals produced by naïve MF. *S* is the number of the elements in the signal **A** and **G**, S=|A|=|G|, where |·| returns the number of vector elements.

Figure 5 shows the results of the filtering accuracy in terms of the number of coefficient images. Note that $T^{2nd}$ and $T^{3rd}$ are the tonal-quantized numbers of coefficient images for the flash and no-flash images, respectively. The filtering accuracy in each case is high overall. For this result, eight coefficient images are enough. This trend is also the same in spatial subsampling, and spatial subsampling is practical because the PSNR accuracy is over 45 dB. It can be seen that the PSNR degradation is more significant when downsampling at the first decomposition. As shown in the next section, downsampling at the first stage has a greater speedup effect; thus, it is up to the application to decide which one to choose.

Figure 6 shows the filtering accuracy of the smoothing parameters. Although the accuracy of DMF varies depending on the parameters, ours has a high accuracy. We can see that it is not very sensitive to changes in spatial parameters $\sigma_{s}$. Each of the two guides has similar range parameters $\sigma_{r1}$ and $\sigma_{r2}$. The smaller the range parameter, the lower the approximation accuracy tends to be, and the effect is more pronounced when the number of decompositions is small. Initial subsampling is also susceptible to this effect. However, the proposed method has an approximation accuracy that is generally better than 45 dB for all parameters, which is sufficient because it becomes difficult for a person to distinguish between two images at around 40 dB [87].

### 7.2. Efficiency Evaluation

We compare the computational time in two cases for efficiency evaluation. One is a comparison between naïve MF and DMF combined with some edge-preserving filters. Another is a comparison with/without subsampling. In this experiment, we apply real-time bilateral filtering (RTBF) [42,43], guided image filtering (GIF) [20] and domain transform filtering (DTF) [33] to (Equation 20). Note that we call DMF with these filters *DMF-Gauss*, *DMF-GF*, and *DMF-DTF*, respectively. Since RTBF can be interpreted as DMF with one guide image for Gaussian filtering, this is equivalent to DMF with two guide images for Gaussian filtering. Therefore, we refer to it as DMF-Gauss. For this experiment, flash and no-flash images were converted to grayscale, and the RGB no-flash image was filtered. In DMF-Gauss, the two-channel images were used as guides; in DMF-DFT and DMF-GIF, the no-flash images were used as guides for DMF, and the flash images were used as guides for DTF and GIF. Note that joint filtering is available for DTF and GIF. Cache-efficient filtering was computed using a one-pass version of Gaussian filtering for DMF-Gauss and box filtering for GIF [64].

Figure 7 shows the processing time results. The processing time of the naïve ML-Guass increases exponentially as the filtering kernel size increases, whereas DMF can be computed in constant-time from Figure 7a. DMF is especially efficient when we use GIF or DTF for the filtering step described in Section 6.1. Furthermore, DMF becomes more efficient by subsampling the spatial domain as shown in Figure 7b. Since DMF and GIF are not decomposable by the proposed method, only a one-step decomposition is possible. Therefore, the computation time for the second decomposition subsampling has not been reported.

### 7.3. Denoising Performance Evaluation

Here, we evaluate the denoising performance of the proposed method; note that it is not the approximation accuracy evaluated by Section 7.1. In our experiments, we used RGB-IR images and simulated RGB-IR fusion by adding noise to the RGB images. Performance is evaluated in terms of PSNR for the noiseless RGB image and the de-noised image; the IR image is not evaluated in terms of PSNR because it is not noiseless and is not the final visible image.

The comparison methods are redundant DCT denoising (DCT) [74], domain transform filtering (DTF) [33], guided image filtering (GIF) [20], cross-field joint image restoration (CFJIR) [88] and high-dimensional kernel filtering (HDKF) [37]. DCT, DTF, and GIF are extended by the proposed method to handle an additional guidance IR image, named DMF-DCT, DMF-DTF, and DMF-GIF. These methods were chosen for their high-speed performance. CGJIR and HDKF already use the characteristics of the guide image; thus, the proposed method extension is ineffective. For evaluation images, we used the RGB-IR dataset, which includes ten images [37] (https://norishigefukushima.github.io/TilingPCA4CHDGF/ (accessed on 16 January 2024)).

Table 1, Table 2 and Table 3 show PSNR results for each method in different noise levels. It can be seen that the classical method, DCT, has the best performance on average for all noise levels due to the DMF extension. CGJIR and HDKF are new dedicated methods for RGB-IR denoising, and performance comparable to these methods has been achieved by extending this method. All DMF extensions also show a steady improvement in performance.

### 7.4. Channel Perfomance Evaluation

Here, we evaluated the denoising effect of the number of channels for flash and no-flash images. In Section 7.1, guide images are grayscaled 2-channel, but here, we use two RGB images, 6-channel. In addition, the number of channels is controlled by using PCA dimensionality compression for guide images [37,89]. Note that the denoising performance is different from the approximation performance. We used a flash/no-flash image dataset [37], which contains ten images. Images are filtered by multilateral filtering. In addition to PSNR, we used structural similarity (SSIM) [90], which is a more robust quality metric. Noise was only added in no-flash images.

Table 4 and Table 5 show the results for each metric. On average, the optimal value is taken by four channels in every metric. The SSIM, which is said to have a high human subjective evaluation value, shows that the value is high enough even for two channels.

Table 6 shows the computational time. The computation time increases exponentially with the number of channels. This indicates that we are suffering from the curse of dimensionality. Therefore, it is better to have as small a number of channels as possible.

### 7.5. Memory Usage Analysis

The memory requirement has linear relations in the number of pixels $N_{p}$. The number of tones $N_{t}$ has exponential relations in the number of channels $N_{c}$ and multiple guidance images $N_{j}$. Consequently, the amount of memory required is $O(N_{p}N_{t}^{N_{c}N_{j}})$, according to Algorithm 1.

The vast memory requirement is one of the limitations, and the limitation is inherited from previous work [42,43]. However, tonal range and spatial domain subsampling can moderate the memory requirement. We can also make memory requirements independent of the number of channels by processing DMF, as discussed in [42,43]. The implementation, however, loses parallelizability and computational efficiency somewhat.

## 8. Multilateral Filtering for Computational Photography

We verify the effectiveness of MF and DMF by applying several applications of sensor fusion in computational photography.

### 8.1. Flash/No-Flash Denoising

Flash/no-flash denoising [16,17] is the representative application for edge-preserving filtering with multiple guidance images. Flashing sometimes causes false edges (e.g., appearance/disappearance of shadow edges), as shown in Figure 8a. Joint bilateral filtering can remove noise in the no-flash image, but it simultaneously preserves the false edges of the flash image (Figure 8c). The conventional method requires multiple steps [17] to solve the problem, while MF requires only one step. As shown in Figure 8d, MF can remove noise while preventing false edges from being transferred. Note that we used the value information in the HSV color space of the no-flash image and the color-flash image as the guidance images. In addition, our algorithm can be efficiently computed by applying efficient edge-preserving filtering, such as domain transform filtering.

### 8.2. Depth Map Refining

Trilateral filtering is effective for refining degraded depth maps by lossy compressing [29]. In the case of a single guidance image, the object boundaries in the depth maps are blurred even if the artifacts are removed; hence, there is a trade-off between denoising performance and the edge preservation effect (Figure 9c). MF can improve this problem by considering both the edges in the depth map and the guidance image (Figure 9d). Although this depth-refinery experiment targets removing coded artifacts, MF can also be applied to noise removal for a depth sensor.

### 8.3. Feathering

We demonstrate the property of our algorithm beyond the Gauss transforms. Guided feathering [20] refines a binary mask for alpha mating near the boundary of the object. For guided feathering, guided image filtering has excellent performance in terms of efficiency and accuracy [20]. The result of naïve guided image filter is shown in Figure 10c. We can confirm that the feather can be computed in detail; however, several noises are simultaneously caused near the object boundary regions. This is because the local linear model of the guided image filter is violated. By contrast, MRGF results hardly include such noises, as shown in Figure 10d, while the detailed feather is computed. This result indicates that MRGF prevents the violation.

### 8.4. Haze Removing

Furthermore, our algorithm with guided image filtering is also effective for haze removal [21]. The large kernel size for filtering is required in haze removal; thus, guided image filtering violates the local linear model, as well as guided feathering. Consequently, some haze remains in Figure 11b. On the contrary, MRGF with guided image filtering can suppress expansion in different objects, as shown in Figure 11c.

## 9. Conclusions

This paper presents an efficient algorithm of edge-preserving filtering with multiple guidance images for sensor fusion signals. Our algorithm, named decomposed multilateral filtering (DMF), can accelerate general multilateral filtering with the Gauss transform and extend various edge-preserving filtering methods to exploit multiple guidance images. In addition, we introduced a method to apply multilateral filtering for the output of multilateral filtering, such as rolling guidance filters [52], named multilateral rolling guidance filtering (MRGF). The experimental results showed that our algorithm has high accuracy and high efficiency. Furthermore, the proposed method is verified by various applications: flash/no-flash denoising, depth map refining, feathering, and Haze removal.

The limitations of our algorithm are that the computational time depends on the image dimensionality and the number of guidance images. However, this problem can be solved by clustering [37,91]. In addition, automatic adjustment of the downsampling amount is also an issue. These issues can be resolved by extending Gaussian KD-trees [47] and permutohedral lattice [48].

## Figures and Tables

**Figure 1 sensors-24-00633-f001:**
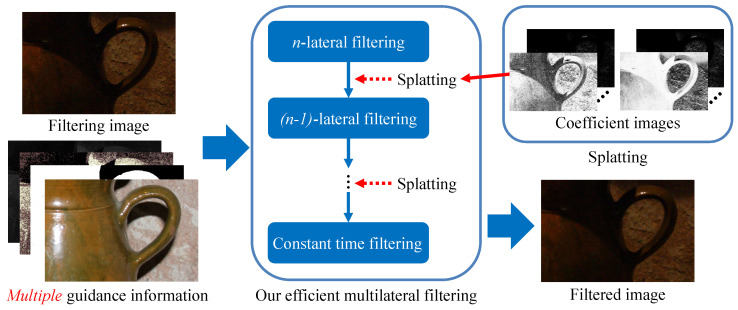
Algorithm overview. *n*-lateral filtering denotes multilateral filtering that multiplies a spatial filter and n−1 range filters. Examples of multiple guidance information are flash images, segmentation masks, and depth maps. The key point of the proposed algorithm is that it decomposes multilateral filtering into a set of constant-time filters. For more information on implementation, our code is available at https://fukushimalab.github.io/dmf/ (accessed on 16 January 2024).

**Figure 2 sensors-24-00633-f002:**
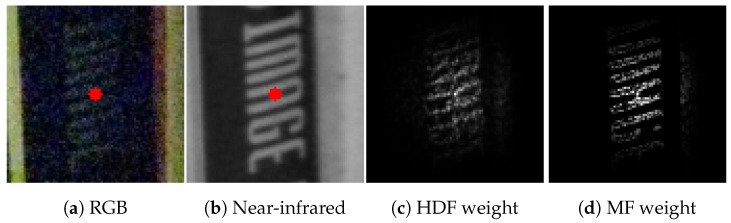
Difference of filtering weights between HDF and MF. HDF weights are computed using the method in [30]. The red point represents the target pixel to compute the kernel weights.

**Figure 3 sensors-24-00633-f003:**
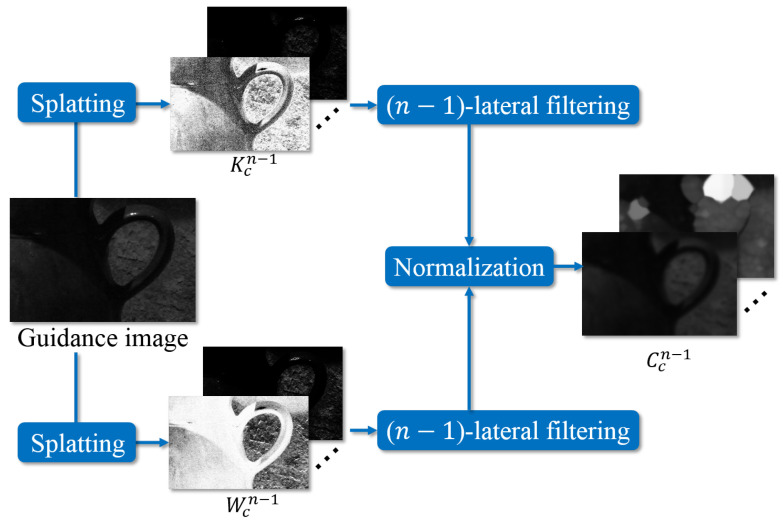
Procedure of splatting in DMF.

**Figure 4 sensors-24-00633-f004:**
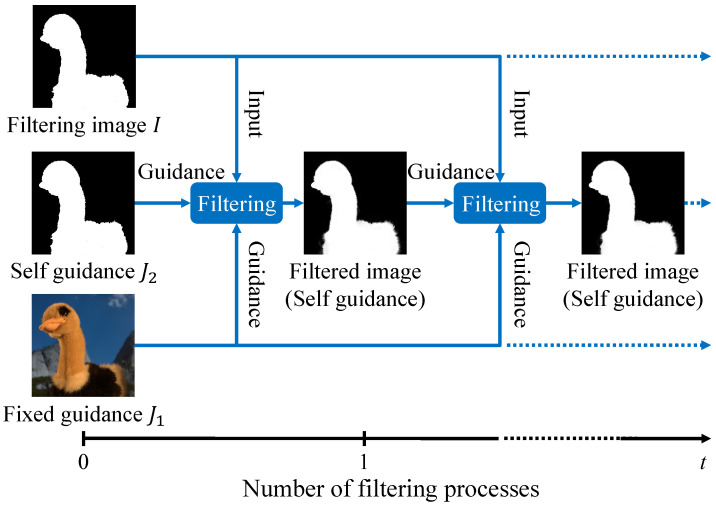
Multilateral rolling guidance filtering. Note that the constraining information $J_{2}$ is the same as the filtering image *I* when the filtering process is the first time.

**Figure 5 sensors-24-00633-f005:**
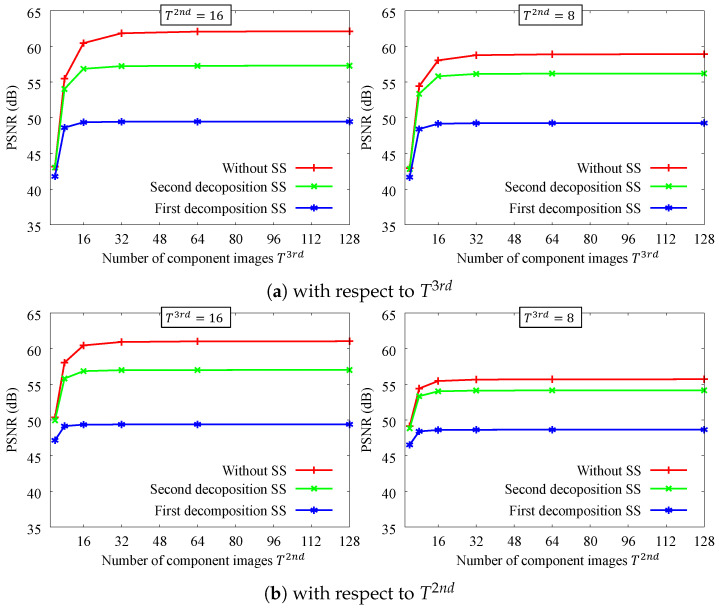
PSNR accuracy with respect to the number of coefficient images. The parameters are $\sigma_{r1}=32$, $\sigma_{r2}=32$, $\sigma_{s}=8$. SS denotes spatial subsampling rates, $\times \frac{1}{16}$. We tested 4 images for the input image.

**Figure 6 sensors-24-00633-f006:**
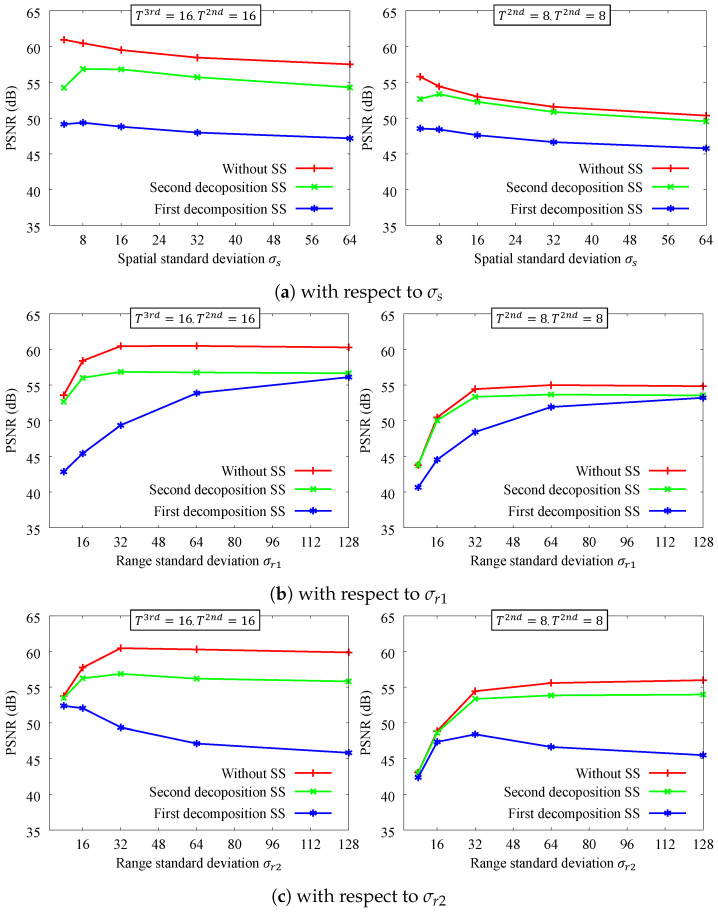
PSNR accuracy with respect to smoothing parameters. (**a**) $\sigma_{r1}=32$ and $\sigma_{r2}=32$. (**b**) $\sigma_{s}=8$ and $\sigma_{r2}=32$. (**c**) $\sigma_{s}=8$ and $\sigma_{r1}=32$. SS denotes the spatial subsampling rate, which is $\times \frac{1}{16}$. The input images are the same as in Figure 5.

**Figure 7 sensors-24-00633-f007:**
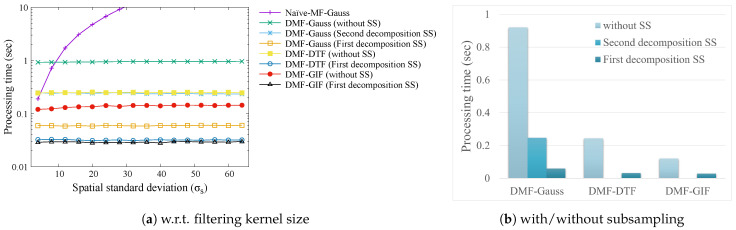
Processing time. The input image size is 1 megapixel (1024 × 1024). The parameters are $T^{3rd}=8$, and $T^{2nd}=8$. SS denotes spatial subsampling rate ($\frac{1}{16}$).

**Figure 8 sensors-24-00633-f008:**
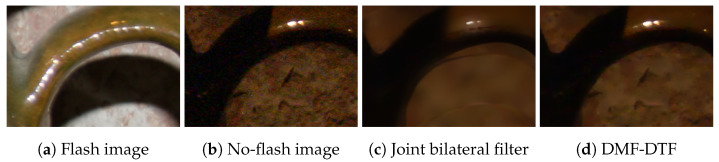
Flash/no-flash denoising without the false edge in the flash image. The parameters are $\sigma_{r1}=64$ (for joint bilateral filtering and DTF), $\sigma_{r2}=16$, $\sigma_{s}=8$, $T^{3rd}=16$.

**Figure 9 sensors-24-00633-f009:**
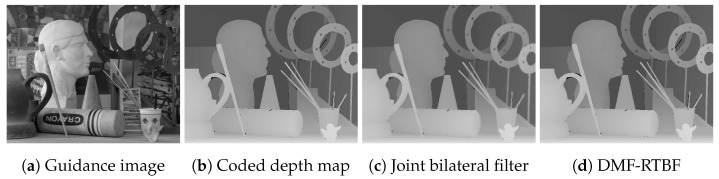
Depth map refining. (**b**) is coded by JPEG (quality factor = 50). The parameters are $\sigma_{r1}=16$, $\sigma_{r2}=16$, $\sigma_{s}=2$, $T^{3rd}=16$ and $T^{2nd}=16$. The values of the ratio of bad pixels [25] (error threshold is 1.0) in (**b**–**d**) are 12.47, 9.24, and 5.38, respectively.

**Figure 10 sensors-24-00633-f010:**
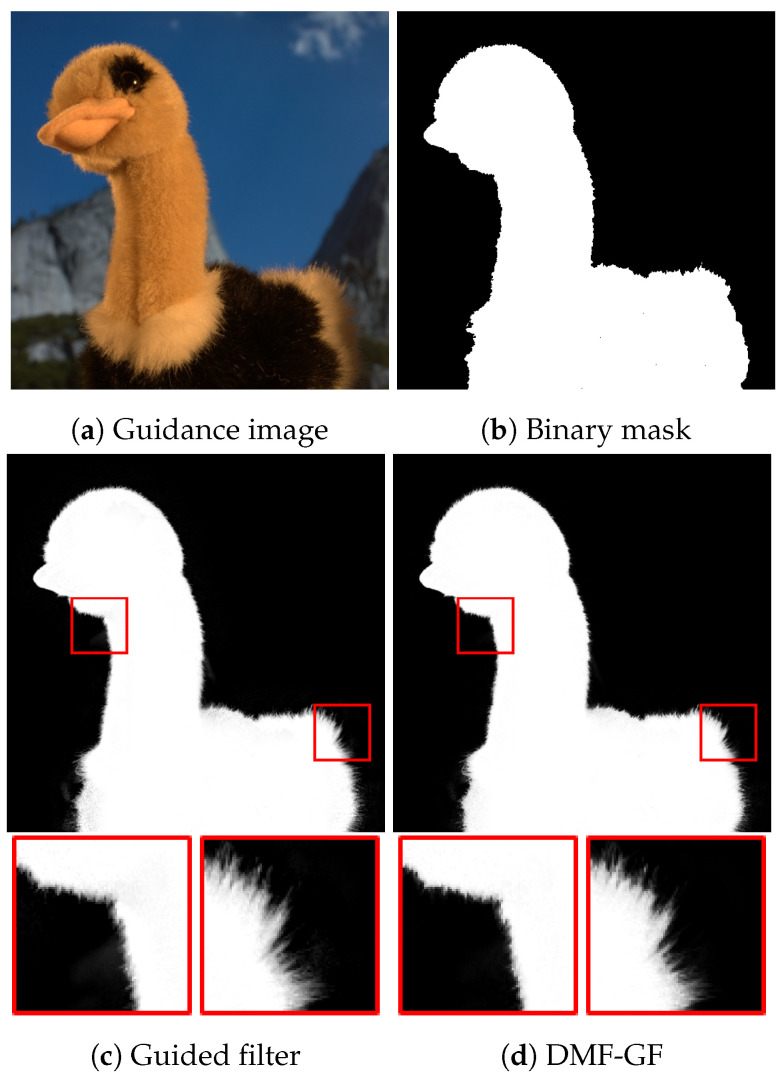
Guided feathering. MRGF’s results are in Figure 4. The parameters are r=20, $\epsilon =10^{-6}$, $\sigma_{r}=160$ and $T^{3rd}=4$. Red boxes indicate magnified areas.

**Figure 11 sensors-24-00633-f011:**
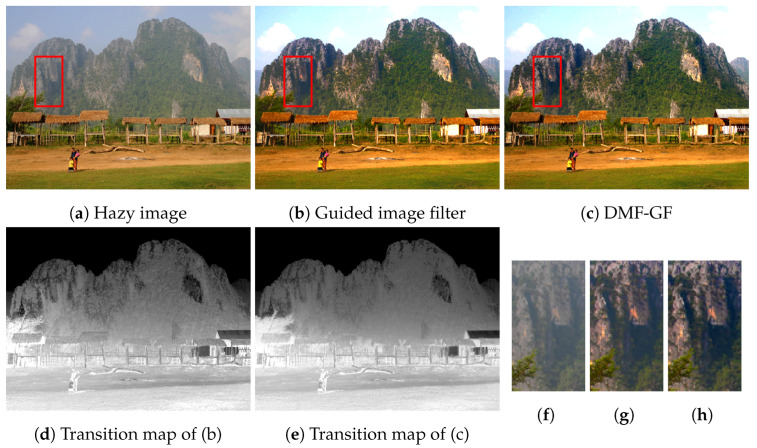
Haze removing. (**f**), (**g**), and (**h**) are the details of (**a**), (**b**), and (**c**) with red boxes, respectively. Our result has been computed by MRGF in Figure 4. The parameters are r=60, $\epsilon =10^{-6}$, $\sigma_{r}=60$ and $T^{3rd}=16$.

**Table 1 sensors-24-00633-t001:** PSNR accuracy of denoising (dB). The Gaussian noise parameter is **σ=10**. * Filters that already use the characteristics of the guide image. Bold numbers mean the best results.

Image	DCT	DMF-DCT	DTF	DMF-DTF	GIF	DMF-GIF	CFJIR *	HDKF *
0	**36.73**	**36.73**	34.41	35.11	34.35	35.26	35.22	35.55
1	33.23	33.41	31.37	31.97	32.34	33.14	32.13	**33.53**
2	**36.16**	36.13	34.50	34.72	34.45	35.10	34.54	35.67
3	39.56	**40.54**	37.53	37.88	37.30	39.38	40.52	40.49
4	36.31	36.47	34.11	34.73	33.90	35.97	35.46	**36.77**
5	**35.51**	35.46	33.76	34.26	33.82	34.68	34.38	34.57
6	**34.06**	33.99	32.21	32.85	32.49	33.25	32.93	33.05
7	36.56	36.69	34.69	35.25	34.60	37.28	37.52	**37.70**
8	**34.50**	34.38	33.04	33.67	33.56	32.99	32.80	33.42
9	36.58	36.82	33.28	34.10	33.77	37.03	37.49	**37.62**
Average	35.92	**36.06**	33.89	34.45	34.06	35.41	35.30	35.84

**Table 2 sensors-24-00633-t002:** PSNR accuracy of denoising (dB). The Gaussian noise parameter is **σ=20**. * Filters that already use the characteristics of the guide image. Bold numbers mean the best results.

Image	DCT	DMF-DCT	DTF	DMF-DTF	GIF	DMF-GIF	CFJIR *	HDKF *
0	32.81	**32.89**	29.90	31.42	29.98	31.91	32.26	32.09
1	29.50	29.92	27.76	28.82	28.28	29.75	29.50	**30.31**
2	31.64	**32.28**	29.47	30.34	29.66	31.44	30.86	31.60
3	36.24	37.34	34.18	34.58	33.45	36.04	**38.59**	36.54
4	32.45	32.85	29.18	31.05	29.09	32.31	32.42	**33.13**
5	31.67	**31.85**	29.56	30.49	29.63	31.02	31.05	31.07
6	30.03	**30.22**	27.50	29.07	27.77	29.90	29.84	29.77
7	32.92	33.27	31.30	32.24	31.03	34.03	**34.98**	34.31
8	30.77	**30.99**	28.56	29.97	29.07	30.08	29.99	30.32
9	32.61	33.21	29.15	30.74	29.32	33.51	**34.47**	34.02
Average	32.06	**32.48**	29.66	30.87	29.73	32.00	32.40	32.32

**Table 3 sensors-24-00633-t003:** PSNR accuracy of denoising (dB). The Gaussian noise parameter is **σ=30**. * Filters that already use the characteristics of the guide image. Bold numbers mean the best results.

Image	DCT	DMF-DCT	DTF	DMF-DTF	GIF	DMF-GIF	CFJIR *	HDKF *
0	30.35	**30.55**	27.24	29.50	27.61	30.21	30.42	30.16
1	27.64	**28.28**	26.03	27.44	26.42	28.27	27.86	28.15
2	28.84	**29.83**	26.08	27.63	26.88	29.12	28.44	28.75
3	33.93	35.01	32.54	33.00	30.82	34.99	**36.41**	34.29
4	30.18	**30.88**	26.28	29.36	26.58	30.63	30.36	30.48
5	29.32	**29.71**	26.79	28.07	27.27	28.48	29.05	29.09
6	27.72	**28.18**	24.65	27.09	25.39	27.88	28.00	27.84
7	30.64	31.19	28.94	30.51	28.82	32.60	**32.85**	31.71
8	28.68	**29.16**	25.98	28.12	26.71	28.31	28.45	28.73
9	30.14	31.22	26.94	29.22	27.18	**32.17**	31.80	30.82
Average	29.74	**30.40**	27.15	28.99	27.37	30.27	30.36	30.00

**Table 4 sensors-24-00633-t004:** PSNR accuracy metrics where **higher** values indicate better flash/no-flash denoising with PCA (dB). Gaussian noise parameter for no-flash images is σ=10. Bold numbers mean the best results.

	Noise	1	2	3	4	5	6
0	28.82	30.92	30.96	31.84	**33.33**	**33.33**	33.32
1	28.47	33.95	34.55	36.34	**36.35**	36.34	36.33
2	28.16	39.91	40.91	**41.32**	41.30	41.29	41.28
3	28.27	37.66	39.56	**39.80**	**39.80**	39.79	39.79
4	28.20	36.34	38.01	38.70	**38.80**	38.76	38.74
5	28.16	38.35	40.43	**40.73**	40.70	40.67	40.66
6	28.18	40.85	42.20	**42.31**	42.29	42.27	42.27
7	28.26	37.44	39.48	39.84	**39.85**	**39.85**	39.84
8	28.28	35.19	37.38	37.52	**37.60**	37.59	37.59
9	28.15	39.35	**41.42**	41.40	41.38	41.36	41.35
Average	28.30	37.00	38.49	38.98	**39.14**	39.12	39.12

**Table 5 sensors-24-00633-t005:** SSIM accuracy metrics where **higher** values indicate better flash/no-flash denoising with PCA. Gaussian noise parameter for no-flash images is σ=10. Bold numbers mean the best results.

	Noise	1	2	3	4	5	6
0	0.932	0.965	0.965	0.967	**0.975**	0.974	0.974
1	0.863	0.970	0.974	**0.984**	**0.984**	0.983	0.983
2	0.723	0.983	0.984	**0.987**	**0.987**	**0.987**	**0.987**
3	0.751	0.973	**0.983**	**0.983**	**0.983**	**0.983**	**0.983**
4	0.769	0.977	**0.985**	**0.985**	**0.985**	0.984	0.984
5	0.693	0.975	**0.985**	**0.985**	**0.985**	0.984	0.984
6	0.650	0.974	**0.984**	0.983	0.983	0.982	0.982
7	0.762	0.976	0.984	**0.985**	**0.985**	**0.985**	**0.985**
8	0.796	0.961	**0.981**	**0.981**	0.980	0.980	0.980
9	0.676	0.973	**0.984**	**0.984**	**0.984**	0.983	0.983
Average	0.762	0.973	0.981	0.982	**0.983**	**0.983**	**0.983**

**Table 6 sensors-24-00633-t006:** Processing time of filtering with multi-channel guide images.

Channels	Time (msec)
1	29.7
2	79.1
3	455.9
4	3380.4
5	27,224.7
6	224,146.0

## Data Availability

Our code is available at https://fukushimalab.github.io/dmf/ (accessed on 16 January 2024).

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
