# Peer review of "Decomposed Multilateral Filtering for Accelerating Filtering with Multiple Guidance Images"

_sensors, 2024, doi:10.3390/s24020633_

Round 1

Reviewer 1 Report

Comments and Suggestions for Authors

1.     The processing times in Figure 7(a) and 7(b) seem to be inconsistent, as can be seen in Figure 7a where the processing time for DMF-DTF is about 0.45s and in Figure 7b where the time is about 0.25s, and in Figure 7b where DMF-DTF and DMF-GF are missing the processing time for the First decomposition SS condition.

2.     The use of PSNR as a criterion for accuracy evaluation in section 6.1 should be marked with a reference.

3.     In section 6.1, the algorithm in the paper should be compared with advanced filters to demonstrate the high accuracy of the algorithm.

Author Response

Thank you for taking time out of your busy schedule to review my paper.

1. The processing times in Figure 7(a) and 7(b) seem to be inconsistent, as can be seen in Figure 7a where the processing time for DMF-DTF is about 0.45s and in Figure 7b where the time is about 0.25s, and in Figure 7b where DMF-DTF and DMF-GF are missing the processing time for the First decomposition SS condition.

I have checked it. As you pointed out, the data did not match well. I re-plotted the data to make it consistent.

The speed is a little faster because I changed the computer at that time. I used a Core i7 7700K (4 cores, 8 threads). The code was also modified to make it faster. The code was also modified throughout to make it faster, because another reviewer pointed out that the experiment had to be completed in a short time. We used more efficient memory and a newer version of OpenMP for parallelization. Guided image filtering is faster because it uses a more efficient method.

The reason why there is no 2nd subsampling in DMF and GIF (GF in the previous manuscript) is that DMF and GIF are not decomposable when interpreted as DMF. The DMF-RTBF in the previous manuscript had 2nd subsampling because RTBF is decomposable when interpreted as DMF. To avoid confusion, we have renamed DMF-RTBF as DMF-Gauss. In addition, naive was renamed ML-Gauss.

Please see the red text in 6.2. Efficiency Evaluation for the changes. Information on the computer used is shown in red text at the beginning of the experimental results section.

2. The use of PSNR as a criterion for accuracy evaluation in section 6.1 should be marked with a reference.

Thank you for your comment. I referred to [a,b] as typical papers using PSNR. Please check it out.

[a] Huynh-Thu, Q.; Ghanbari, M. Scope of validity of PSNR in image/video quality assessment. Electronics Letters 2008, 44, 800–801(1). https://doi.org/10.1049/el:20080522.

[b] Hore, A.; Ziou, D. Image quality metrics: PSNR vs. SSIM. In Proceedings of the International Conference on Pattern Recognition (ICPR), 2010, pp. 2366–2369. 

3. In section 6.1, the algorithm in the paper should be compared with advanced filters to demonstrate the high accuracy of the algorithm.

Thank you for your comment. It is difficult to include other methods as approximations because 6.1 is the section that evaluates the approximation accuracy as PSNR. Therefore we make two subsections to evaluate our method: Sec 6.3 and 6.4.

In Sec 6.3, we compare newer methods of cross-field joint image restoration (CFJIR) and high-dimensional kernel filtering (HDKF) with the proposed method of DMF for three filters: DCT denoising, domain transform filtering, and guided image filtering. The proposed method can extend any spatial filtering to have additional edge-preserving factors; thus, we can extend the existing method. These 3 filters are chosen by the reason of the its speed. We show that our method is more effective than the competitive method, and the proposed DMF can improve the performance of existing filters. The proposed method is not effective because CFJIR and HDKF already actively use guided images.

In Sec 6.4, we verify the effectiveness of the proposed method for multichannel effect to improve the proposed method, and PCA is combined to control the number of channels. Experimental results show that better control of the number of channels provides greater control over accuracy and speed.

Reviewer 2 Report

Comments and Suggestions for Authors

This paper presents an edge-preserving filtering algorithm with multiple guiding images for sensor fusion signals. This algorithm can speed up general multi-way filtering using Gaussian transform and extend various edge-preserving filtering methods to use multiple navigation images. The proposed method has been tested by various applications. The experimental results showed that the algorithm has high accuracy and high efficiency.

The article may be published in the journal “Sensors”.

Author Response

Thank you for your interest. We have revised the paper to make it better.
Thank you for taking time out of your busy schedule to review the paper.

Reviewer 3 Report

Comments and Suggestions for Authors

The paper is refreshingly presenting a technique to achieve edge-preserving filtering that does not rely on the use of AI. It makes profit from a sensor fusion scheme, and the framework of multilateral filtering, to propose a novel approach called decomposed multilateral filtering. The approach aims to reduce the computational complexity commonly associated with multilateral filtering.

I kindly suggest the authors to provide greater detail regarding the use cases in which a technique such as the one presented in the paper (or multilateral filtering, for that matter). Particularly, given that it requires alternative sources of image information to achieve the edge preserving filtering.

Also, it would behoove the scientific soundness of the article to have the results compared against other state-of-the-art techniques in image denoising and feature-preserving image denoising, some of which the authors have already cited in the bibliography.

Comments on the Quality of English Language

The paper is well written, and clearly organized. I did not detect any problems or limitations that should be addressed for a final submission. 

Author Response

The paper is refreshingly presenting a technique to achieve edge-preserving filtering that does not rely on the use of AI. It makes profit from a sensor fusion scheme, and the framework of multilateral filtering, to propose a novel approach called decomposed multilateral filtering. The approach aims to reduce the computational complexity commonly associated with multilateral filtering.

Thank you for taking time out of your busy schedule to review the manuscript.

1. I kindly suggest the authors to provide greater detail regarding the use cases in which a technique such as the one presented in the paper (or multilateral filtering, for that matter). Particularly, given that it requires alternative sources of image information to achieve the edge preserving filtering.

Thank you for pointing that out. We have created a section on related work in Section 2 to clarify the position of this study with respect to image fusion. Our contribution to image fusion is to propose an extension of weighted average processing, which is a commonly used tool for image fusion. The proposed method is expected to play many roles, as it extends existing average and weighted average methods to edge-preserving filters guided with additional signals while preserving their own filtering properties. Experiments show several applications as examples.

2. Also, it would behoove the scientific soundness of the article to have the results compared against other state-of-the-art techniques in image denoising and feature-preserving image denoising, some of which the authors have already cited in the bibliography.

Thank you for your comment. We make two subsections to evaluate our method: Sec 6.3 and 6.4.

In Sec 6.3, we compare newer methods of cross-field joint image restoration (CFJIR) and high-dimensional kernel filtering (HDKF) with the proposed method of DMF for three filters: DCT denoising, domain transform filtering and guided image filtering. The proposed method can extend any spatial filtering to have additional edge-preserving factors; thus, we can extend existing method. These 3 filters are chosen by the reason of the its speed. We show that our method is more effective than the competitive method and the proposed DMF can improve the performance of existing filters. The proposed method is not effective because CFJIR and HDKF already actively use guided images.

In Sec 6.4, we verify the effectiveness of the proposed method for multichannel effect to improve the proposed method, and PCA is combined to control the number of channels. Experimental results show that better control of the number of channels provides greater control over accuracy and speed.

Reviewer 4 Report

Comments and Suggestions for Authors

This paper proposes an efficient algorithm for edge-preserving filtering with multiple guidance images. The structure of the proposed paper is clear which is easy to read and understand, but there are still some points which should be improved.

1     Please check English, which should be improved.

2     How to filter images from different modal sensors with the proposed method in this study?

3     How does the number of gradient images affect the filtering performance?

4     How to deal with redundant information between gradient images, which can affect the efficiency of the proposed algorithm?

5     How to select the spatial subsampling rates for blurred images of different degrees?

Comments on the Quality of English Language

This paper proposes an efficient algorithm for edge-preserving filtering with multiple guidance images. The structure of the proposed paper is clear which is easy to read and understand, but there are still some points which should be improved.

1     Please check English, which should be improved.

2     How to filter images from different modal sensors with the proposed method in this study?

3     How does the number of gradient images affect the filtering performance?

4     How to deal with redundant information between gradient images, which can affect the efficiency of the proposed algorithm?

5     How to select the spatial subsampling rates for blurred images of different degrees?

Author Response

This paper proposes an efficient algorithm for edge-preserving filtering with multiple guidance images. The structure of the proposed paper is clear which is easy to read and understand, but there are still some points which should be improved.

1. Please check English, which should be improved.

Thank you for pointing that out. I have edited the entire manuscript using Grammaly and Writefull, both of which are paid editing tools. Please check it out.

2. How to filter images from different modal sensors with the proposed method in this study?

The proposed method extends any existing spatial filter and allows filtering by adding weights from different guide images. Signals with different modalities have different signal characteristics, and performance can be improved by filtering while taking into account the effects of different signals. For example, an RGB signal and an IR signal can be filtered to produce an image with a high signal-to-noise ratio by taking advantage of the IR signal's high dynamic range characteristics. The proposed method extends the existing spatial filtering denoising method to use IR signals as guiding information, enabling high-performance denoising based on IR signal characteristics.

3. How does the number of gradient images affect the filtering performance?

4. How to deal with redundant information between gradient images, which can affect the efficiency of the proposed algorithm?

Thank you for your comment. This method does not use gradient information, but I will read it as the number of guides. In addition, 3 and 4 will be answered collectively.

The effect of the number of guides on the proposed method is difficult to generalize since it depends on the signal characteristics, but the closer the signal characteristics are, the greater the effect. For example, we consider a six-dimensional signal with two RGB signals for flash and no-flash images. Denoising this signal can be effectively filtered by using all 6 dimensions.

However, as you pointed out, the signal has redundancy. In such cases, dimensional compression processes such as PCA are often used to eliminate redundancy and reduce the number of dimensions. We have added an experiment to measure the effect of the number of signals by changing the amount of dimensionality compression.

In the case of flash/no-flash denoising, performance peaks at 4 channels in most cases. Any more than four channels will either degrade performance or reduce the effect due to the noise in the data. The number of guide images has a significant impact on computation time, and dimensional compression, such as PCA, helps speed up the proposed method.

For more detailed information, please refer to the Experiments section.

5. How to select the spatial subsampling rates for blurred images of different degrees?

The subsample rate depends on the signal and the blur spread. This paper does not show how to automatically determine the subsampling rate, but experiments (Section 6.1) have shown that the lower the approximation order, the less the effect of downsampling, and that the larger the σr to downsample, i.e., the less effective the downsampling is without a filter with high edge preservation effect. The experiment shows that the lower the approximation order, the less the effect of downsampling.

There are also methods to automatically determine the downsample, such as Gaussian KD-Tree and permutohedral lattice, but these will be discussed in a future issue. These issues were added as a limitation to the method at the end.